# Cross-Talk between the Cytokine IL-37 and Thyroid Hormones in Modulating Chronic Inflammation Associated with Target Organ Damage in Age-Related Metabolic and Vascular Conditions

**DOI:** 10.3390/ijms23126456

**Published:** 2022-06-09

**Authors:** Ljiljana Trtica Majnarić, Zvonimir Bosnić, Mario Štefanić, Thomas Wittlinger

**Affiliations:** 1Department of Family Medicine, Faculty of Medicine, Josip Juraj Strossmayer University of Osijek, Huttlerova 4, 31000 Osijek, Croatia; ljiljana.majnaric@gmail.com; 2Faculty of Dental Medicine and Health Osijek, Josip Juraj Strossmayer University of Osijek, Crkvena 21, 31000 Osijek, Croatia; 3Department of Medicinal Chemistry, Biochemistry and Clinical Chemistry, Faculty of Medicine, Josip Juraj Strossmayer University of Osijek, Huttlerova 4, 31000 Osijek, Croatia; mstefanic@mefos.hr; 4Department of Cardiology, Asklepios Hospital, 38642 Goslar, Germany; dr.wittlinger@gmx.de

**Keywords:** aging, chronic inflammation, metabolic-inflammation cross-talk, cardio-metabolic disorders, cytokine IL-37, thyroid hormones

## Abstract

Chronic inflammation is considered to be the main mechanism contributing to the development of age-related metabolic and vascular conditions. The phases of chronic inflammation that mediate the progression of target organ damage in these conditions are poorly known, however. In particular, there is a paucity of data on the link between chronic inflammation and metabolic disorders. Based on some of our own results and recent developments in our understanding of age-related inflammation as a whole-body response, we discuss the hypothesis that cross-talk between the cytokine IL-37 and thyroid hormones could be the key regulatory mechanism that justifies the metabolic effects of chronic tissue-related inflammation. The cytokine IL-37 is emerging as a strong natural suppressor of the chronic innate immune response. The effect of this cytokine has been identified in reversing metabolic costs of chronic inflammation. Thyroid hormones are known to regulate energy metabolism. There is a close link between thyroid function and inflammation in elderly individuals. Nonlinear associations between IL-37 and thyroid hormones, considered within the wider clinical context, can improve our understanding of the phases of chronic inflammation that are associated with target organ damage in age-related metabolic and vascular conditions.

## 1. Introduction—An Association between Chronic Inflammation and Age-Related Metabolic and Vascular Conditions

Inflammation constitutes an essential response of tissues to infection or injury [1]. The phases and time-dependent evolution of acute inflammation are well known. The key event is leukocyte migration from small blood vessels to the site of infection or injury, followed by vasodilation and plasma leakage, which in turn augments the inflammatory reaction. A short time after the inflammatory response begins, the program to resolve inflammation ensues, with the aim to repair damaged tissue [2]. To allow many coordinated actions to flow smoothly, phases of acute inflammation involve a rapid switch in biosynthetic processes and cell turnover, including proliferation, differentiation, and programmed cell death (apoptosis) [1,2]. When the intensity of a local inflammatory reaction is higher, some systemic reactions also emerge, including mobilization of inflammatory and immune cells from the bone marrow, synthesis of acute-phase proteins in the liver, activation of the neuroendocrine stress axis, and changes in metabolic pathways [1,3]. These systemic reactions are only temporary and have protective roles. The proinflammatory cytokines released at the site of inflammation send a danger signal to the whole body [4]. If an inflammatory stimulus is very strong, or an inflammatory reaction dysregulated, it creates conditions for the host immune response to fail and multiple organ damage to develop, which is termed sepsis [5].

The body of knowledge is growing on inflammation as a major mechanism underlying the development of chronic health conditions, the prevalence of which increases in older age (>60), such as for type 2 diabetes (T2D), atherosclerosis, heart disease, osteoporosis, neurodegenerative diseases, and cancer [6]. As postulated by the currently most prevalent theory of aging, “inflammaging,” changes in the innate immune system that occur as a response to life-long exposure to external and internal antigenic stimuli, and an excess of adipose tissue associated with obesity and age-related changes in body shape, are the main source of chronic, low-grade inflammation [7]. In light of that, aging burdened with chronic diseases can be viewed as accelerated aging, as both “successful” and “unsuccessful” courses of aging are likely to use the same mechanisms, but at different rates [8]. The impact of environmental factors, such as unhealthy lifestyles, obesity, and chronic psychological stress, are likely to be more important than the impact of genetics in accelerating the aging processes [8,9]. If, however, defense mechanisms are preserved until advanced age, this may prolong survival and maintain older individuals in a good functional state [10].

It is now well established that different types of stimuli apart from microbial components, termed pathogen-associated molecular patterns (PAMPs), can trigger innate immune receptors, thus activating proinflammatory pathways [1,7]. A wide range of these sterile inflammatory signals, termed danger-associated molecular patterns (DAMPs), have been identified so far. Apart from molecules released from dying immune cells or damaged tissue and the breakdown products of the extracellular matrix (EM), DAMPs also include metabolic intermediates produced in excess in conditions associated with obesity, metabolic syndrome (MS), and T2D, such as free fatty acids (FFAs), diacyl-glycerol, advanced glycation end-products, and lipids enriched in ceramides and phospholipids [11,12]. In addition to these signals, conditions in the microenvironment associated with glucose deprivation and changes in the metabolic pathways that can induce an increase in the production of reactive oxygen species (ROS) were also found to augment inflammatory signals. If aging is associated with obesity, this may provide additional sources of inflammation. In these terms, activated macrophages, which infiltrate adipose tissue, and senescent adipocytes have been recognized as an important source of proinflammatory cytokines [7]. Recent evidence also highlights the role of disturbances in the composition of the intestinal microbiome, which is known to accompany aging and metabolic conditions associated with obesity, in providing microbial signals, which can maintain the innate immune system in a continuously active state [13].

The central role within the cell in sensing and integrating various kinds of stress signals has been given to inflammasomes, the molecular complexes that serve as the second messenger, by translating the totality of the cell’s stress signals into the innate immune responses [14,15]. Increased production of proinflammatory cytokines is known to induce insulin resistance (a decreased efficacy of insulin on glucose utilization in insulin-sensitive tissues, including muscles, adipose tissue, the liver, and the brain), which exacerbates metabolic disturbances, further increasing the level of inflammation [14,15]. Finally, the brain responds to increased systemic levels of inflammation by dysregulating the neuroendocrine feedback loops, the aim of which is to reverse the disrupted tissue homeostasis [16]. However, if activated in the long term, these adaptation mechanisms act to promote further changes in metabolic and inflammatory pathways. At the organ and tissue levels, there is a vicious cycle of metabolic, inflammatory, and vascular disorders, operating within the common pathophysiological background that links related disorders together, including obesity, hypertension, MS, T2D, and cardiovascular disease (CVD), resulting in the progression of target organ damage and atherosclerotic vascular disease [8,17,18].

Many separate mechanisms involved in generating inflammatory responses as well as mechanisms that operate at the interface of inflammatory and metabolic disorders have been identified so far. However, knowledge is missing on the phases of chronic inflammation associated with organ damage in age-related metabolic and vascular conditions and on mechanisms that regulate the dynamics of organ-damage change. There is a consensus among researchers that we need a more integrated view of how metabolic and vascular disorders develop and progress over time. Recent evidence indicates that the cytokine IL-37 represents the main tissue protective factor. It is well established that dysregulation in the hypothalamic–pituitary–thyroid (HPT) axis accompanies age-related metabolic disease and CVD. Recent evidence links this dysfunction with increased levels of inflammation. Taking these pieces of evidence together with some of our own observations, we aimed in this work to elaborate the hypothesis that these two regulatory mechanisms, the one mediated by the cytokine IL-37and the one mediated by thyroid hormones (THs), are likely to be the main compensatory mechanisms by which tissues respond to inflammation-mediated disruption in metabolic homeostasis.

## 2. Chronic Inflammation Associated with Target Organ Damage in Age-Related Metabolic and Vascular Conditions

Only little is known about factors that can trigger and sustain inflammatory responses in target organs and about immune cells infiltrating tissues and organs in different phases of organ damage [1]. The main reason is that real-life studies of this kind are difficult to perform in humans, and animal models can only provide information on selected mechanisms [1,2]. Nevertheless, the currently available evidence indicates that, in T2D-associated target organ damage, innate immune receptors and inflammasomes, notably including the inflammasome NLRP3, and increased production of pro-inflammatory cytokines, such as tumor necrosis factor-α (TNF-α), interleukin-1β (IL-1β), and IL-6 [11,14], play a role here. Apart from this, noninflammatory residential cells have also been recognized as likely contributors to tissue-related inflammation and remodeling [2,6]. These cells, under inflammatory conditions, acquire the capacity to produce cytokines and are otherwise actively involved in processes associated with increased cell apoptosis and tissue fibrosis, which are known to accelerate organ remodeling and damage.

It has long been known that target organ disease in patients with T2D is closely related to development of macrovascular and microvascular complications, including entities such as ischemic (atherosclerotic) heart disease, congestive heart failure (CHF), stroke, chronic kidney disease (CKD), peripheral arterial disease, and retinopathy [19]. Recent evidence indicates that the brain and the skin are target organs in MS and T2D, too [20,21]. The mechanisms proposed to underlie cognitive impairments include either small cerebral vessel disease, as observed in the vascular type of cognitive impairment, or neurodegeneration, as in Alzheimer’s disease [21,22]. The current understanding of mechanisms underlying structural and functional changes of the components of the cardiovascular (CV) system, including large- and medium-sized arteries, the heart, and the kidneys, is that they are the result of metabolic, oxidative, and hemodynamic stress, as an effect of a long-term action on cells and tissues of this system, of overlapping CV risk factors, such as hypertension, obesity, and T2D, synergized with the effects of stochastic and senescent processes [23,24]. Some particular mechanisms, which, by acting simultaneously, may lead to dysfunction of the microvasculature and development of increased stiffness of large arteries, include decreased availability of the vasodilator substance nitrite oxide, hyperproduction of ROS due to mitochondrial dysfunction, macromolecule modification by glycation and oxidation, activation of the renin-angiotensin system (RAS), increased activity of the tissue type angiotensin II (ANG II), residential cell switching on the senescence-associated secretory phenotype, loss of autophagy (a mechanism of cell waste management), and impairment of protein homeostasis (proteostasis) [24,25,26,27].

In parallel to the progression of fibrotic and structural changes of the CV system, hemodynamic disorders are aggravated, leading to the vicious cycle of hemodynamic and metabolic changes and microcirculation dysfunction, with target organ disease being the final result [23,27]. Microcirculation has a pivotal role in maintaining metabolic and hemodynamic homeostasis of the tissue due to its role in optimizing the supply of nutrients and oxygen to the tissues, in removing tissue waste, and in buffering detrimental effects of fluctuations in hydrostatic pressure, which occurs at this very end of the vascular network [23,27]. Impairments in the functioning of the microvasculature may contribute to aggravation of hypertension and metabolic disorders, leading to the progression of target organ disease, and finally, to increased risk of macrovascular CV events [23,27,28]. In this latter effect, microcirculation plays a critical role by increasing adhesion and activating blood leukocytes on the surface of its dysfunctional endothelium. When taking into account the enormously large surface of the microcirculation, this part of the vascular system is considered the main driving force in promoting inflammation in the CV system, ultimately contributing to the progression of atherosclerosis in large arteries.

However, the initial reaction of tissues of the CV system to metabolic, hemodynamic, and hypoxic noxious challenges is not associated with inflammation; it initially involves the adaptive responses of residential tissue cells to these challenges, but without the flow of the blood leukocytes from the circulation to tissues [1,23]. When the tissue stress level reaches a certain threshold, inflammation is likely to develop. How this occurs in a real-life setting is not completely understood. As suggested by the available evidence, the composition of inflammatory and immune cells that migrate to tissues of the target organs might be different from cells known to participate in the atherosclerotic process in large arteries, mostly including macrophages, dendritic cells (DCs), NK cells, CD8+ (cytotoxic) T lymphocytes, and Th1 lymphocytes [28,29]. In target organ damage, a pivotal role is attributed to the subset of CD4+ T lymphocytes, termed T helper 17 (Th17) lymphocytes [30,31]. This lymphocyte subset produces cytokines of the IL-17 family (IL-21, IL-22, IL-26, and IL-17A and F), which act to create an inflammatory environment [32]. A large role is attributed to the cytokine IL-17A in tissue damage and remodeling due to its effects in mobilizing macrophages and neutrophils and inducing the tissue type Ang II [32]. Although neutrophils have been traditionally viewed as protagonists of an acute inflammatory response, recent evidence indicates that they participate in chronic inflammation in the context of target organ damage associated with age-related metabolic and vascular conditions [33]. Neutrophils exert devastating effects through their ability to release a large amount of proinflammatory cytokines, ROS, and proteolytic and oxidizing enzymes into the surrounding tissues. The tissue type Ang II acts synergistically with IL-17A to promote tissue inflammation and remodeling via mechanisms such as immune cell mobilization, ROS production, and promotion of tissue fibrosis [34]. In an inflamed microenvironment, other immune cells and activated residential cells may also produce IL-17A, thus augmenting inflammation [35]. In inflamed tissue, the presence of cytokines IL-6, IL-21, IL-23, IL-1β, and the growth factor, termed “the transforming growth factor beta” (TGF-β), and factors indicating tissue hypoxia, such as hypoxia-inducible factor 1 (Hif-1), create a microenvironment suitable for differentiation of the naïve CD4+ T cells to proinflammatory Th17 cells, whose hallmark is increased expression of the transcription factor RORγt [31]. The Th1 cell-specific cytokine interferon gamma (IFN-γ) also participates in promoting Th17 cell differentiation, thus helping establish the Th1/Th17 proinflammatory pathway. This pathway is controlled by tissue T regulatory (Treg) cells—the T-cell subset that differentiates from T naïve cells when there is a sufficient dose of TGF-β in the microenvironment. The hallmark feature of these cells is high expression of the transcription factor Foxp3. The cytokines secreted by Treg lymphocytes, TGF-β and IL-10, exert immunosuppressive effects and are involved in tissue fibrosis and remodeling [31].

Overall, the primary role of the inflammatory response in chronically inflamed tissue is to restore tissue homeostasis by employing tissue repair programs [1]. Apart from residential macrophages that, when induced by IFN-γ, acquire an ability to secrete a large amount of proinflammatory cytokines, such as IL-6, IL-1β, and TNF-α, the inflamed microenvironment supports the differentiation of macrophages that can initiate the tissue-repair process via their ability to secrete cytokines with anti-inflammatory properties, such as IL-4, IL-10, and IL-13 [36]. When the proinflammatory cytokines prevail over the anti-inflammatory cytokines, notably including a sufficient amount of IL-6, it is likely to turn the balance between Th17 and Treg cells towards the Th17-dependent proinflammatory pathway becoming the predominant one [31]. In general, the Th17–Treg balance is highly sensitive to variations in environmental cues [37]. Apart from changes in cytokine profiles, changes in metabolic conditions can also shift this balance in either of the directions, as dictated by the metabolic requirements of the two main, albeit opposite, Th-related pathways. In this regard, the Th17 lymphocyte subset, a short-lived proinflammatory T-cell subset, usually employs glycolysis to generate energy and substance and maintain stable differentiation, and this subset is more dependent on FFA synthesis and on supply with the amino acid glutamine [37]. In contrast, Treg cells depend more on oxidative phosphorylation and FFA oxidation than on other metabolic pathways and employ the mevalonate pathway for proliferation.

The proinflammatory tissue environment, as found in metabolic and vascular conditions associated with aging, was shown to promote the shift in the Th17–Treg balance towards the Th17 signaling pathway becoming the predominant one [31]. Increased secretion of proinflammatory cytokines, along with an increase in oxidative stress, may in turn exacerbate inflammation, which aggravates metabolic and vascular disorders [38]. As the conflict between Th17 and Treg cells is never fully resolved, like Th17 cells, Treg cells can also expand to some extent, forming the basis for continuous tissue remodeling and fibrosis [29,36].

## 3. Thyroid Hormone Modes of Action in Aging and Age-Related Diseases

Thyroxine (T4) and triiodothyronine (T3) are collectively referred to as THs [39]. As the major prohormone, T4 is produced and secreted into the blood by the thyroid gland in a process tightly regulated by the HPT axis. The thyroid gland also provides a small fraction of physiologically active T3; the remaining fraction is locally generated from the peripheral 5′-monodeiodination of blood-borne T4 by the enzymes iodothyronine-deiodinase (DIO) 1 and 2 [40]. The third DIO isoform, DIO3, is a hypoxia-sensitive enzyme that inactivates T4 and T3 to reverse (r)T3 and 3,5-diiodothyronine (T2), respectively [41]. The uptake and efflux of THs by target cells is facilitated by several membrane transporters [42]. Inside the cell, canonical T3 signaling executes its function on gene regulation by binding THs to nuclear TH receptors (THRs) [39,40]. This step is mediated mainly by interaction of THs with coregulators and requires either a THR/THR homodimer or a heterodimer of THR bound with RXR to interact with hormone-responsive DNA elements in target genes [43]. The THRs can also interact with other nuclear hormone receptors, such as peroxisome proliferator-activated receptors, retinoic acid receptors, and liver X receptors [44], which allow for binding to a wide repertoire of nucleotide sequences that contribute to mitochondrial biogenesis and proteostasis [45], cell energetics, and different metabolic pathways, including cholesterol, glucose, and fatty acid turnover, in different tissues [46,47]. Notably, the increased mitochondrial generation of ROS is a side effect of the TH signaling [48], providing a parallel route to oxidative DNA damage and premature senescence [49]. Impaired aerobic and redox metabolism may in turn regulate TH responsiveness in different tissues: the increased transcriptional activity mediated by T3 after oxidative inhibition of PTP1B [50], a phosphatase that regulates translational repression of THRAP1, a subunit of the Mediator complex that controls transcription by THs and other nuclear hormone receptors [51], provides the most recent evidence for this. Another example is NADH-dependent redox regulation of SIRT1, a longevity-linked histone deacetylase that, among other roles, acts as a direct coactivator of THRβ1, thus promoting tissue-specific conservation of energy [52].

Apart from these genomic effects, several cytosolic and membrane-locked thyroid hormone-binding proteins have been reported to mediate nongenomic, transcriptionally independent actions of THs [53]. Examples include μ cristallin, which mediates cytosolic retention and nuclear delivery of T3 [54], and αvβ3 integrin, which harbors two binding domains, one activating PI3K via Src kinase and the other regulating MAPK1 and MAPK2 [55]. Nongenomic actions of THs also involve components of the NOS3-PKGII, SREBP1, SIRT1, AMPK, CaMKKβ, PI3K-AKT-FOXO1, mTOR-p70S6K, and ATM/PRKAA pathways, which further modulate transcription by integrating nutrient and redox inputs [43,56]. As a result, THs regulate energy expenditure via the central nervous system and through a direct role in the major metabolic tissues, such as the liver, skeletal muscle, and adipose tissue. In this process, profound, simultaneous effects on lipophagy, mitophagy, lipogenesis, lipid oxidation, redox balance, cholesterol turnover, gluconeogenesis, pancreatic insulin secretion, peripheral insulin resistance [57,58], fibrogenesis [59], and inflammation [60] have been reported, providing a range of effects on cellular and systemic homeostasis that can comprise protective or detrimental responses during aging. The net effect, however, often depends on the context.

A general view of the TH-dependent signaling and metabolic pathways under aging conditions is as follows: At any given time, more than 99% of circulating THs are bound to plasma proteins; the remaining unbound fractions constitute a bioavailable form and are referred to as free (f)T4 and fT3 [40]. In contrast to plasma fT4/fT3 partitioning, estimating this ratio in the extravascular space is not as straightforward. For example, the situation may be different in aging tissues, where alterations in tissue architecture progressively accumulate over time due to age-associated tissue remodeling. Broadly, a fibrotic matrix forms progressively, whereby senescent, secretory, and proinflammatory cells accumulate in many tissues, often in conjunction with profound immune alterations [61]. The age-associated inflammation increases vasopermeability and extravasation of plasma constituents, changing the ability of THs to cross the endothelial barrier and access target tissues [41,62,63]. Thus, the exact fraction of fT4/fT3 that goes into the extracellular matrix and reaches cellular targets is likely to depend on the environment. In addition, hypoxic and malignantly transformed tissues exhibit pathologic vascularization and leakages, leaving open the question of how THs are transported across the endothelium in such different beds. Conversely, THs may affect the endothelial barrier and capillary recruitment in their own right, thus influencing the transfer to their target tissues [41]. This effect of THs is controlled by angiogenic genes, which are under direct control of THs via αvβ3 integrin receptors on the plasma membrane [55]. The evidence further indicates that the transport efficiency of THs across the vascular beds might vary with endothelial expression of TH transporters [42]. This, especially in the hypothalamus and anterior pituitary, may impair the central negative regulation of thyrotropin (TSH)-releasing hormone (TRH) and TSH secretion, mostly by modulating local T3 levels. The activity of this axis is primarily controlled by the release of TRH from neurons localized in the hypothalamic paraventricular nucleus to portal capillaries, facing chemosensitive DIO2+ tanycytes from the median eminence (ME), a structure at the base of the hypothalamus [64,65]. Notably, hypothalamic microinflammation emerges in early stages of aging and MS [64], changing the hypothalamic neurochemistry and landscape of supporting cells that regulate neuronal maturation and maintenance [66,67]. The rich cellular landscape of the ME is not spared from inflammaging. Microglia, brain-resident macrophages, take on a primed proinflammatory phenotype in ME together with vascular and leptomeningeal cells (VLMCs), which belong to a family of highly related barrier-forming fibroblasts [67]. Both VLMCs and microglia take part in immune-related pathways during aging, contributing to an asynchronous propagation of senescence in aged tanycytes. How these processes might differ in successful and pathologic aging is open to investigation. Other important sources of interindividual variations in serum TH concentrations are the profound changes in body shape and composition, excessive or reduced calorie intake, and altered intestinal microbiome in aged individuals, all of which may provide feedback to the HPT axis [68,69,70]. For example, high-normal TSH levels have been associated with obesity [70]. Thyroid dysfunction, both hypo- and hyperthyroidism, has been associated with increased risk of CVD [71].

## 4. The Role of the Cytokine IL-37 in Regulating Acute Inflammation

The cytokine IL-37 is a member of the IL-1 family of cytokines, otherwise known for its pivotal role in promoting inflammation [72]. This cytokine (a former member 7 of the IL-1 family) was recently characterized by computational cloning, which revealed its role as a negative regulator of the cytokine IL-18, which in addition to IL-1β is the key proinflammatory cytokine of the IL-1 cytokine family [73]. There is growing interest in the therapeutic potentials of IL-37 in T2D and CVD [74]. To better understand its modes of action, we need to elucidate the role of innate immunity in acute inflammation and initiation of the adaptive (T cell-mediated) immune response.

### 4.1. Innate Immunity and Acute Inflammation

As a result of the fact that IL-1β and IL-1α, the first members of the IL-1 cytokine family to be discovered, were found to have proinflammatory properties, the whole IL-1 cytokine family was thought to have a role in amplifying innate immune and inflammatory responses. Technological advances in the last few decades have enabled the discovery of new cytokines of this family, with both pro- and anti-inflammatory properties [75]. It has been further determined that the IL-1 receptor (IL-1R) family members not only have ligand-binding chains, but also contain molecules that act as negative regulators of cellular signaling [76]. Today, after all molecules of the IL-1 family have been identified and classified according to the unique nomenclature, we can better understand the principles of functioning of this signaling system. Owing to its strong proinflammatory properties, this system also possesses mechanisms of self-control and fine-tuning of proinflammatory effects, the aim of which is to preserve tissues at the site of infection (inflammation) from unnecessary collateral damage and autoimmune reactions [77]. Molecules of the IL-1 cytokine family, which function to maintain control over unrestrained inflammation, include soluble molecules, such as IL-1 and IL-36 receptor antagonists (IL-1Ra and IL-36Ra), suppressor cytokines IL-37 and IL-38, and receptor-inhibiting molecules, such as IL-18 binding protein (IL-18BP) [74,77]. In addition, there are many checkpoints along the IL-1-dependent intracellular signaling pathways where variations in substrate concentrations or changes in component switch or cleavage can serve as limitation factors and can inhibit propagation of the proinflammatory signals [77].

In principle, when cytokines IL-1β or IL-1α bind to their IL-1R1 expressed on the surface of innate immune cells such as macrophages and antigen-presenting DCs, proinflammatory signals are initiated within these cells [72]. IL-1R1 is a ligand-binding receptor molecule. After ligation to its agonist, this molecule undergoes structural changes, which allows the coreceptor molecule IL-1R3 to approach and form the functional tripartite receptor complex. Intracellular signaling proceeds via intracellular domains of these two receptor components, which are structurally similar to intracellular domains of the Toll-like innate immune receptor and are therefore termed Toll IL-1R (TIR) domains. When TIR domains approach each other, conditions are created for binding to the adaptor molecule MyD88, whereby this molecule becomes phosphorylated and capable of initiating the canonical activation signal to intracellular kinases (IRAKs) [72,75]. This is an essential proinflammatory pathway mediated by IL-1β as the prototypic proinflammatory cytokine and involves the proteins TRAF6 and TAK1 and IKK kinases, leading to activation of the transcription factor nuclear factor kappa B (NF-κB) [72]. This transcription factor is known for its pivotal role in regulating a large array of genes involved in inflammatory and immune responses (Figure 1) [78]. Among the most important of these genes are those that code proinflammatory cytokines, for example, tumor necrosis factor alpha (TNF-α), IL-1α, IL-β, IL-6, IL-12, IL-15, and IL-18; those that code chemokines (inflammatory cell chemoattractant factors), such as IL-8, monocyte chemoattractant protein-1 (MCP-1), and granulocyte- and granulocyte-macrophage colony stimulating factor (G- and GM-CSF); those that code adhesion molecules, such as intercellular adhesion molecule 1 (ICAM-1); those that regulate the survival, activation, and differentiation of innate immune cells and T lymphocytes; and those that participate in inflammasome regulation (the intracellular multi-protein activators of caspases—enzymes involved in lytic programmed cell death or pyroptosis) (Figure 1) [72,78,79,80]. Thus, the resulting cellular activities include gene transcription, translation processes, autophagy (degradation of damaged or redundant cellular components through lysosome-dependent regulated mechanisms), pyroptosis (a caspase-dependent inflammation-mediated lytic cell death), necroptosis (a caspase-independent inflammation-mediated lytic cell death), immune-mediated metabolic changes, and oxidative phosphorylation of enzymes participating in signaling pathways (Figure 1) [77].

The role of the IL-1 family of molecules, in fact, is to amplify or fine-tune the innate immune responses on antigens recognized by innate immune receptors. These receptors, such as Toll-like receptors (TLRs), are displayed on the surface of innate immune cells, including mainly macrophages, DCs, and neutrophils, or are positioned intracellularly, in the proximity of inflammasomes (the intracellular sensing system), such as NOD-like receptors (also termed nucleotide-binding domain and leucine-rich repeat pyrin-containing protein receptors) (NLRs) (Figure 1) [73,74,75,76,77,79,80,81]. By using the innate immune receptors, innate immune cells search the microenvironment for the presence of PAMPs or DAMPs. As molecules of the IL-1 and TLR families share similar functions in providing innate immune responses, these two signaling systems share the same group of cytosolic domains—TIR domains [72,77]. This way, by using the prepared intracellular signaling components, the proinflammatory cytokines of the IL-1 family can amplify inflammatory signals initially generated by TLRs (Figure 1) [77].

There is another important inflammation amplification pathway with prolonged effects. It involves new transcriptions of the precursor molecules of the proinflammatory cytokines IL-1β and IL-18 and inflammasome receptor component NLRP3 (Figure 1) [72,77]. Several types of inflammasomes have been identified in humans, but the one containing the NLRP3 receptor has been examined best, as it plays a pivotal role in human physiology [79,80]. In addition to the receptor component, inflammasome complexes also comprise an adaptor protein ASC and an executioner component—an enzyme from the caspase family, notably, caspase-1. The role of the NLRP3 complex is to process the precursor proinflammatory cytokines IL-1β and IL-18 into mature, more active forms. The execution program involves activating the proteolytic enzyme caspase-1 and is tightly controlled by the canonical reaction, which starts with the assembly of particular NLRP3 inflammasome components and is followed by the recruitment and activation of pro-caspase 1, finally leading to proteolytical modification of the IL-1β and IL-18 precursor forms [77,80,81]. Once activated, caspase-1 also proteolytically modifies the membrane protein gasderin-D, which then dimerizes, forming large pores in the cell membrane through which the mature forms of the cytokines IL-1β and IL-18 are expelled from the cell into the extracellular space (Figure 1) [82]. Formation of the pores may eventually cause cell membrane rupture and cell lysis, leading to the release of large quantities of cytokines and other inflammatory active molecules from the cell. Additional ways by which the proinflammatory cytokines amplify inflammation involve processes such as the dispersion of activation signals to other, nonimmune cell types, including endothelial, epithelial, and mesenchymal cells, and the recruitment of new inflammatory and immune cells to the site of inflammation via dysfunctional endothelial lining of the microcirculation (Figure 1) [1]. Variations in cytokine concentrations at the site of inflammation, which develop over time, and different cytokines acting in concert within the cytokine network, are mechanisms by which the proinflammatory cytokines fine-tune the phases of acute inflammatory reaction, which evolves from rarefaction of the noxious stimulus, via activation of the specific immune response, to heal damaged tissue (Figure 1) [1]. During an acute inflammatory response, the cytokine IL-18 acts synergistically with other proinflammatory cytokines of the IL-1 family, using the same downstream signaling pathways to enhance production of these proinflammatory cytokines [72,83]. In the course of this pathway, IL-18 initiates the signaling cascades by binding to its heterodimeric receptor, which consists of the ligand-binding chain IL-18Rα and the signal-transduction chain IL-18Rβ. Upon ligation of IL-18 to the ligand-binding chain, these two parts of the receptor approximate, allowing for the trimeric complex to assemble, which is then capable of initiating downstream signaling [83]. The specific role of this cytokine is to induce the cytokine IFN-γ, whose role is to activate macrophages and generate the cytotoxic, cell-mediated immune response. In addition to macrophages, it involves NK and Th1 cells, too, and is essential for eliminating intracellular pathogens and aberrant and damaged cells (Figure 1) [84,85]. The signaling pathway engaged in producing IFN-γ takes the TRIF adaptor molecule-dependent pathway, which employs the endocytosed TLRs and induces IFN-γ synthesis via activation of the IFN-γ-inducing transcription factor IRF3 (Figure 1) [78,80]. Given the self-aggressive potential of cell-mediated immunity, strictly regulated IL-18-dependent signaling is essential for preventing uncontrolled tissue damage. In physiological conditions, however, the activity of the cytokine IL-18 is mainly regulated by IL-18BP, which binds this cytokine with higher affinity than the ligand-binding unit of the IL-18 receptor and thus suppresses the effect of this cytokine in generating the Th1-mediated immune response [84,85]. A rise in the production of the proinflammatory cytokines is an obligatory precondition for initiating the specific (T cell-mediated) immune response [85]. Namely, if present in the microenvironment as a sign of danger, the proinflammatory cytokines provide an activation signal to DCs by upregulating the costimulatory molecules and peptides of the major histocompatibility complex (MHC) on their surface. This step is necessary so that DCs can acquire the capability to prime naïve CD4+ Th lymphocytes to differentiate into one of the effector Th cell subsets, including Th1, Th2, and Th17 cell subsets (Figure 1) [85].

### 4.2. Suppression of Acute Inflammation by the Cytokine IL-37

The broad spectrum of suppressor activities of the cytokine IL-37 relies on the fact that this cytokine acts at strategic points in the proinflammatory pathways. Its mode of action primarily involves suppressing the effects of the proinflammatory cytokine IL-18, and thus, indirectly, also the effects of IL-1β [73]. The cytokine IL-37 induces its effects by binding to the IL-18Rα receptor component, which then attracts the decoy receptor IL-1R8 instead of the signal-conducting component IL-18Rβ (Figure 2). The newly developed IL-37/IL-18Rα/IL-1R8 receptor complex reduces IL-1β and IL-18-dependent signaling and can also reduce TLR-dependent signaling by sequestering the adaptor molecule MyD88. This results in dumping activities of the major intracellular proinflammatory pathways, including the IKK- and MAPK-dependent pathways, which are needed to control synthesis of the proinflammatory cytokines and to keep the PIK3/Akt/mTOR pathway active, whose role is to control the metabolic effects of inflammation (Figure 1) [73,86]. Although signaling pathways of the cytokine IL-37 have not been completely clarified, it seems that, in addition to dumping the proinflammatory pathways, IL-37 also acts to actively promote the anti-inflammatory pathways, operating via the negative signal molecules, such as STAT3, Mer, and PTEN (Figure 2) [73,87]. Apart from operating via the cell-surface receptors, IL-37 may also translocate to the nucleus, where it binds to the nuclear DNA and suppresses transcription of the proinflammatory genes (Figure 2). To be endowed for this nuclear activity, the IL-37 precursor molecules must be processed into the mature forms via proteolytic enzyme caspase-1 activity. The mature molecules can enter the nucleus after forming complexes with the molecule Smad 3—a kinase operating in the pathway of the immunosuppressive cytokine TGF-β (Figure 2).

Emerging evidence indicates that IL-37 is constitutively expressed in human cells at low levels but is upregulated by inflammatory stimuli [73,86,87]. When intracellular concentrations of the mature form of this cytokine reaches some threshold, their precursor molecules are exported from the cell to the extracellular space. The extracellular proteases (for example, released from activated neutrophils) are thought to process the IL-37 precursors into the mature forms, which then may exert the extracellular functions (Figure 2). By modulating inflammation and the innate immune response, IL-37 also affects the adaptive (T cell-mediated) immune response. One of the proposed mechanisms is suppressing the stimulation of naïve Th lymphocytes by DCs, which is possible via reduced expression of costimulatory molecules such as CD40 and CD86 and MHC molecules on the surface of DCs. Another important mechanism is promoting the development of Treg cells, whose role is to limit inflammation and tissue damage by suppressing the generation of effector T-cell subsets (Figure 2) [88].

## 5. The Cytokine IL-37 as the Key Regulator of Chronic Inflammation Associated with Organ Damage in Age-Related Metabolic and Vascular Conditions

### 5.1. Immune-Metabolic Disturbances in Age-Related Metabolic and Vascular Conditions

As recent evidence suggests, inflammation and metabolism are inextricably linked at both the molecular and system levels and should not be explored separately [89]. Regarding cells of the immune system, their metabolic status is a critical determinant of their function [90]; on the other hand, while executing different functions in inflammation and host defense, immune cells adjust their metabolic pathways and usually need an additional supply of some nutrients in order to function and proliferate [91]. When viewed from the whole-body perspective, chronic inflammation has been recognized as an independent risk factor for the development and progression of age-related metabolic and vascular disorders, such as obesity, hypertension, T2D, and CVD [6,7,8]. Conversely, the presence of metabolic disorders, as we have already stated above, can accelerate aging via the release of a wide range of molecules that serve as danger signals and can become augmented [11,12,14,92]. In addition, intestinal dysbiosis usually accompanies obesity and T2D, representing a large source of microbial signals; in this way, and owing to insufficient synthesis of some essential nutrients, this condition may contribute to persisting inflammation [13,93]. A better understanding of mechanisms that operate at the crossroads between the innate immune system and metabolic disorders could enable us to develop improved therapies for curing chronic inflammatory and metabolic disorders. In this regard, it has been recognized that the same proinflammatory intracellular signaling pathways are used irrespective of the types of challenge, whether in response to infection or when driven by metabolic cues or increased tissue damage, as in the presence of chronic diseases [11,12,14,90]. In this context, inflammasomes play the key role through their effects in integrating proinflammatory signals and modulating metabolic pathways, mostly by controlling production of the proinflammatory cytokines (Figure 1) [94]. Accordingly, increased activity of the inflammasome NLRP3 has been found in metabolic and CV conditions [14,22]. Two distinct processes that operate at the crossroads of metabolic and proinflammatory pathways and that are of the utmost importance for the development and progression of age-related metabolic and vascular conditions deserve to be described in more detail. They include (1) trained immunity and (2) insulin resistance.

### 5.2. Trained Immunity

Until recently, immunological memory (an ability of the immune system to mount a more efficient defense reaction in repeated encounters with the same microorganism) was considered to be an exclusive feature of the adaptive (T cell-mediated) immune system [85]. A growing body of evidence, however, indicates that long-term adaptive changes may also affect monocytes/macrophages, resulting in their enhanced responses to repeated stimulation with infectious and noninfectious challenges [95]. As indicated by the results of transcriptional and epigenetic studies, those genes whose activity is affected in trained immunity are the ones involved in immune functions and in stabilizing glycolytic metabolic pathways [96]. Such studies have revealed the key mechanisms underlying trained immunity, including rewiring of cellular metabolism and induction of post-translational histone modifications (epigenetic changes). These mechanisms result in increased chromatin accessibility for inflammatory stimuli and a long-term increase in production of the proinflammatory cytokines. An essential step in the process of epigenetic reprogramming is the switch in cell energy metabolism from oxidative phosphorylation to glycolysis [36,91]. This process is regulated by activating the Akt/mTOR/Hif pathway, resulting in increased production of lactate and disruption of the tricarboxylic acid cycle (TCA), also known as the Krebs cycle (Figure 3). The purpose of these metabolic changes is to meet the requirement of activated immune cells to rapidly generate adenosine triphosphate (ATP), the energy-storage molecules needed to execute immune cell functions and synthesize new components [91,97,98]. If Krebs cycle activity decreases and some alternative metabolic pathways are activated, the intracellular concentrations of some metabolites increase, for example, citrate, succinate, and fumarate. Increased availability of these metabolites within the cell were found to accelerate histone modifications and the development of epigenetic changes (Figure 3) [36,91]. Apart from the effects of epigenetic reprograming, changes in intracellular concentrations of some metabolites, by-products of glycolytic metabolic pathway activation, especially including increased concentrations of lipid moieties such as FFAs, cholesterol, and cholesterol derivatives or some amino acids, may also have a role in changing signal transduction processes via mechanisms such as changes in composition of the cell membrane and other cell structures or by directly influencing the signaling pathways [91,97,98].

### 5.3. The Complex Molecule mTOR-Mediated Regulation of the Effector T-Cell Commitment

The serine threonine kinase, termed mammalian target of rapamycin (mTOR), has emerged as the key regulator of immune functions (Figure 3) [97]. The assembly of mTOR with different adapter proteins may form two different variants with different functional abilities. The variant mTORC1 has an essential role in committing naïve Th cells to differentiate into the effector Th1 and Th17 cell subsets, whereas mTORC2 signaling regulates differentiation of the Th2 cell subset [97,98]. The kinase mTORC1 is activated through signaling of the PI3K-Akt pathway and, in cells, has a role in regulating essential functions such as metabolism, protein synthesis, proliferation, and survival by sensing and integrating information on the availability of nutritional and growth factors. In T cells, activation of mTOR has a central role in regulating T-cell differentiation [98].

In general, when T cells are in the resting state, as is the case for naïve Th cells, they employ the catabolic mode of cellular metabolism, using autophagy to secure amino acids for protein synthesis and mitochondrial oxidative phosphorylation to maintain energy production [98,99]. This quiescent state is actively maintained, being controlled by regulatory transcription factors such as FOXO1, which was found to promote expression of inhibitory proteins, for example, PI3K inhibitor phosphatase and tensin homolog (PTEN) [100]. Upon activation, in contrast, T cells turn on the anabolic metabolic mode by switching from regular metabolism based on the Krebs cycle to prevalent participation in the cell metabolism of glycolysis; this switch is associated with increased nutrient uptake and biosynthetic activities. Activation of the complex mTORC1 is needed to sustain glycolytic metabolic reprograming [98]. The critical requirement, which regulates the activity of mTORC1, is sufficient supply of the cell with branched-chain amino acids, such as leucine and glutamine. An increase in the intracellular ATP/AMP ratio as a consequence of the switch of T cells to glycolytic metabolism suppresses activation of the AMP-kinase (AMPK), thus protecting mTORC1 from inhibitory activity of this kinase [98,101]. During T cell proliferation, the cell glucose transporters GLUT1, GLUT3, and GLUT4 are also induced and anchored to the cell membrane to promote glucose utilization [98].

It is important to mention that effector T cells, including cytolytic CD8+ T cells and Th1, Th2, and Th17 CD4+ effector cell subsets, are highly dependent on glycolytic metabolic reprogramming and that memory T cells and Treg cells use low levels of glucose and primarily utilize FFAs for oxidation and maintaining energy homeostasis (Figure 3) [98,99]. The activated PI3K-Akt signaling pathway upstream of mTORC1 increases the expression of glucose transporters, thus enhancing glycolysis and skewing the Th17/Treg balance [31,99].

A growing body of evidence indicates that Treg cells are equipped with some degree of plasticity, which is driven by oscillations in metabolic programming [31,99,100]. In the proliferation state, such as when Tregs recognize antigens and then migrate to the site of infection, Tregs regulate energy metabolism through the mTORC1-dependent pathway; otherwise, by turning on TORC1-independent metabolic pathways or oxidative phosphorylation, these cells regain a suppressor function. Taken together, mTORC1 signaling is necessary for Treg cells to proliferate and establish a suppressor function by promoting cholesterol/lipid metabolism. In the next steps of Treg cell generation, activation of the PIK3/Akt signaling pathway needs to be fine-tuned and well dosed in order to maintain Treg cell renewal and function.

In conditions when nutrients such as glucose or energy levels are low, AMPK activation, as the key cellular energy sensor, exerts an inhibitory effect on the activity of mTORC1, thus negatively regulating ATP-consuming biosynthetic processes and upregulating FFA oxidation and substance supply by autophagy (Figure 3) [31,97,101]. The production of IFN-γ is especially sensitive to glucose availability, and AMPK acts to suppress IFN-γ mRNA translation under conditions of glucose deprivation [101]. Proliferating T effector cells may eventually continue to upregulate the TCA cycle if there is a sufficient supply of amino acid glutamine to the cell by engaging glutamine to produce pyruvate, the main energy source for the TCA cycle [99]. By responding to a decreased amount of oxygen in the cell, the transcription factor Hif1 becomes activated and induces glycolytic enzymes, enabling promotion of Th17 cell differentiation [31]. Thus, the role of HifF-1 is to sustain the persistence/survival of Th17 cells in conditions associated with tissue hypoxia, such as in severely inflamed tissue [31,99]. The stability of the established Th17/Treg balance, under certain metabolic conditions, is maintained by epigenetically modifying RORγt and Foxp3 expression. Many specific metabolites that alter the tolerance/inflammation balance by upregulating either FOXP3 or RORγt expression have been identified [99]. A tolerogenic response is promoted, for example, when short-chain FFAs are processed by commensals in the colon [93]. This usually occurs in situations when an individual is in good health and in the absence of metabolic disturbances such as obesity or T2D [102].

### 5.4. Insulin Resistance

Under normal physiological conditions, insulin acts via insulin receptors (IRs) and stimulates glucose utilization to produce energy in insulin-dependent tissues (skeletal muscle, adipose tissue, the liver, pancreas, and the brain) [103]. The effects of insulin on metabolism extend to promoting glycogen and protein synthesis in muscle, lipogenesis in adipose tissue, and glycogen and FFA storage in the liver, via its inhibition effect for gluconeogenesis and glycogenolysis. In short, activation of IRs leads to tyrosine phosphorylation of insulin receptor substrate-1 (IRS-1), which, via activation of the PI3K-Akt signaling pathway, mediates glucose transport and other insulin-dependent effects. An intact PI3K-Akt signaling pathway is therefore critical for maintaining tissue homeostasis, as it mediates vital cellular processes, such as oxygen-dependent energy production, growth, proliferation, and survival. Impairments in this pathway induce insulin resistance (reduced tissue responsiveness to the physiological action of insulin), obesity, and T2D in insulin-sensitive tissues [104,105]. Conversely, metabolic disturbances associated with obesity often exacerbate insulin resistance by modulating this pathway, which, via excessive insulin production in the pancreatic β-cells, leads to a vicious circle of glucose homeostasis impairment, ultimately resulting in hyperglycemia and the development of T2D [103].

The process of activating the PI3K-Akt pathway is initiated when activated PI3K phosphorylates its substrate phosphatidylinositol biphosphate in intracellular membranes, thereby recruiting signaling proteins such as Akt. In fact, Akt becomes fully activated through two phosphorylation processes, including that on the threonine residue and that on the serine residue. Once activated, Akt regulates processes such as translation of GLUT4 and cellular energy production by stimulating glycolysis. At the same time, Akt reduces gluconeogenesis and FFA oxidation by inhibiting the protein FoxO1 and glycogen synthase kinase 3 (GSK3) and reduces protein synthesis by inhibiting mTORC1, whereas it increases cholesterol and FFA accumulation by activating the sterol regulatory element-binding proteins (SREBPs). The molecule PTEN is known to be the main negative regulator of PIK3 [103].

Dysfunctions in the PI3K/Akt pathway, by increasing insulin resistance and impairing glucose transport and glycogen synthesis, play a crucial role in the development of obesity and T2D. In contrast, obesity- and T2D-related disorders, such as increased production of FFAs and intracellular lipid deposition, mitochondrial dysfunction and increased oxidative stress, and increased production of proinflammatory cytokines, may promote insulin resistance by acting at different points of the PI3K/Akt pathway, especially in skeletal muscle, where the majority of insulin-stimulated glucose utilization takes place [104]. These disorders may promote insulin resistance by activating stress kinases (instead of Akt kinases), such as the c-Jun N-terminal kinase (JNK) family of MAPK kinases, and IKK kinases, which act to inhibit insulin-mediated signal transductions by triggering the inhibitory serine phosphorylation of IRS-1.

When this process is viewed from a more global perspective, it can be said that obesity-related disorders create an inflammatory microenvironment associated with increased intracellular formation of inflammasome complexes, notably including the NLRP3 type of inflammasome and thus amplifying both the innate immune response as well as the immune response locally, in tissue, and at the system level [94,106]. A growing body of evidence indicates that activation of inflammasomes by DAMPs plays a crucial role in obesity-induced inflammation and development of insulin resistance [106,107]. Moreover, the well-established view today is that an inflammatory state induced by the excess of nutrients and metabolic disorders, termed metainflammation, together with the chronic low-grade inflammation that accompanies aging, termed inflammaging, is the key driving force in the course of developing chronic diseases and functional deficits in older individuals [108].

### 5.5. The Role of the Cytokine IL-37 in Reversing Immune-Metabolic Disturbances Associated with Chronic Inflammation

After reviewing the most important processes at the crossroads of metabolic disorders and chronic inflammation, the multiple roles of the cytokine IL-37 in reversing these disorders have become easier to understand. The dual role (extracellular and intracellular) of this cytokine in suppressing production of proinflammatory cytokines, mostly including IL-1β and IL-18, can explain its role as the major natural inhibitor of immune responses in chronic inflammatory and autoimmune diseases, including cancer [109]. Moreover, recent evidence indicates that the biological effects of this cytokine extend beyond direct suppression of proinflammatory cytokines and also involve its effects on cell metabolism. The most important metabolic effect of IL-37 is in activating the MAPK and suppressing the Akt/mTORC/HIF-1 signaling pathways, which may reverse the established bias from mainly glycolysis to predominantly the participation of oxidative phosphorylation in cell metabolism [88,109].

Closely related to its effect in modulating the metabolic pathways, this cytokine is also involved in reversing the effects of the aberrant activation of trained immunity, as is the case in chronic inflammatory conditions (Figure 2) [109]. This effect of IL-37 was confirmed in a number of experimental models. It has been shown, for example, that treating experimental animals with IL-37 may reduce the expression of epigenetic markers of trained immunity or reverse metabolic changes, characteristics of trained immunity. In studies using human cell cultures extracted from individuals suffering from chronic health conditions, IL-37 was shown to inhibit inflammatory mediator production. In aged experimental animals, transgenic expression of IL-37 was shown to restore the function of both CD4+ and CD8+ T lymphocytes, which indicates its role in reversing the effects of chronic inflammation on age-related decline in immune responses [110]. Treatment of experimental animals with human recombinant IL-37 was found to reverse the metabolic costs that are often incurred in systemic inflammation and improve oxidative respiration and exercise tolerance in experimental animals [111]. The metabolic effects of IL-37, such as the effect on improving sensitivity to therapy with insulin in patients with T2D, could be, even partly, caused by the effect of IL-37 in reversing intestinal dysbiosis [112]. In a series of experiments, IL-37 has also been shown to play a pivotal role in fibrotic processes, associated with tissue and organ remodeling. Actually, the results of these experiments indicate that the lack of IL-37, either for genetic reasons or due to inappropriate upregulation in tissues as an expression of failed homeostatic regulatory mechanisms, is associated with increased tissue fibrosis [113,114]. Thus, in patients with idiopathic pulmonary fibrosis, a progressive and destructive lung disorder of unknown origin, the amount of IL-37 was found to be lower in alveolar epithelial cells and alveolar macrophages than in healthy controls [114]. As shown by in vitro experiments, mechanisms by which IL-37 may attenuate fibrotic processes are probably manifold, including decreased expression of mRNA and fibronectin and collagen synthesis in fibroblasts, inhibition of fibroblast proliferation on stimulation with TGF-β, and improved autophagy in fibroblasts. As indicated by experimental studies, variations in IL-37 concentrations can modulate inflammatory signals and the magnitude of tissue inflammatory responses to environmental challenges by being positioned at strategic places, such as in alveolar or intestinal epithelial cells and alveolar macrophages, that is, at the interface of the mucosal immune system and the outer environment [115].

## 6. Cross-Talk between the Cytokine IL-37 and Thyroid Hormones—A Relationship between Tissue-Related and Whole-Body Responses to Stress

THs are key regulators of energy and nutrient metabolism in many tissues. Recent evidence has revealed that THs may exert their effects by using a large array of mechanisms, including classical, genomic, and alternative; nongenomic signaling pathways; engagement with different types of nuclear receptors and transcription factors; and post-translational modifications of THRs. Alternatively, THs may act indirectly by activating metabolic pathways or by regulating inflammasomes and autophagy, or fine-tuning gene transcription via the influence of selecting certain miRNA (as reviewed in Section 3). This new evidence has expanded the diversity of genes and functions that can be regulated by THs. Apart from their known role in controlling metabolic rates and energy expenditure, in physiological and different nutritional conditions, recent evidence accentuates the role of THs in inflammation, fibrotic processes, and microcirculation remodeling, whereby all processes play a role in creating tissue architecture [55,56,59,60]. Taken together, it can be said that THs are a part of the fine-tuned regulatory network aimed at maintaining the balance between tissue energy expenditure and storage, responding to actual tissue metabolic needs and phases of remodeling.

This homeostatic network is often dysregulated in aging and cardiometabolic conditions. In support of this statement, both peripheral metabolism and action of THs, and central regulation of TH production, are altered under aging conditions (reviewed in Section 3). As studies on animals have suggested, metabolic activity diverges among organs and tissues [116]. A higher variability in serum TSH has been observed in the older population than in younger age groups, as demonstrated by widening serum TSH range, even in a euthyroid state [117,118]. In general, aging is associated with lower thyroid function, as suggested by the fact that subclinical hypothyroidism and nonthyroidal illness syndrome (NTIS), as distinguished from autoimmune thyroid diseases, are the most prevalent thyroid disorders in older individuals [117,119]. Under the term subclinical (latent) hypothyroidism, we consider a discrete low thyroid function marked with an only minor elevation in TSH, whereas THs are within the reference range. The term NTIS indicates a state characterized by low serum T3 and increased rT3, but without an increase in TSH, which can be seen in severe, acute conditions or in chronic conditions associated with multimorbidity and frailty, notably including CHD and CKD [71,119]. A widely accepted view that explains the hypothyroid state in aging is that a reset in HPT cross-talk, seen in the elderly, is an effect of reduced basal metabolism. Alternatively, this is considered to be a protective homeostatic reaction, aimed at preventing the elderly from extreme catabolism while favoring “physiological aging” [119].

Here, we defend the hypothesis that the hypothyroid state in older individuals is the system’s compensatory mechanism for chronic inflammation in target organs, which takes place when local tissue compensatory mechanisms are not able to overcome the tissue-related metabolic costs of inflammation. This may develop in situations in which IL-37, as a major protagonist of maintaining tissue homeostasis under inflammatory conditions, fails to reverse disturbed tissue homeostasis. Actually, when stimulated by increased tissue inflammation, this cytokine acts to decrease proinflammatory pathways and reverse the metabolic costs of inflammation (as reviewed in Section 4 and Section 5). Although the exact pathways and checkpoints through which THs and IL-37 may act in synergism under inflammatory conditions to preserve metabolic tissue homeostasis are not clearly defined and probably depend on context, we hold that inflammation-mediated insulin resistance is the main stimulus for homeostatic alignment of both THs and IL-37. In this respect, obesity and MS have been strongly associated with TSH in numerous observational studies [58]. Treating patients with T2D with metformin, an oral hypoglycemic drug that improves insulin resistance, was found to reduce TSH levels both in overt and subclinical hypothyroidism, but not in euthyroid patients [120]. Metformin acts by activating AMPK and SIRT1, energy-dependent metabolic cell sensors involved in maintaining cell cycling under stress conditions [52,120]. Included among the diverse roles of SIRT1, a NAD-dependent histone deacetylase, in maintaining body homeostasis are also those involved in cytokine production, epigenetic changes associated with trained immunity, and hormonal regulation of energy metabolism via homeostasis regulation in the HPT axis [121,122,123,124]. The links between insulin resistance, IL-37, and TSH under aging conditions should therefore be searched within those circuits involving regulation via oxidative metabolism and SIRT1, but accounting for differences in cell and tissue specificities and phases of tissue remodeling [122].

In this respect, THs are known to exert their metabolic actions mainly in metabolically active tissues, thus coupling with the action of insulin in glucose-dependent energy-producing metabolic pathways [103,104]. Under physiological conditions, this action involves transformation of T4 to T3 by DIO1 and DIO2 and binding of T3 to its nuclear THRs to regulate transcription of genes included in these pathways [39,40]. By the mechanism of selecting certain miRNAs, activation of this classical TH signaling pathway also keeps intracellular production of ROS and the NLRP3 inflammasome activation under control, thus promoting anti-inflammatory conditions [60]. Increased inflammation, which develops under conditions of aging and the presence of cardiometabolic disorders, in contrast has an impact on suppression of insulin-mediated activation of the PI3K/Akt metabolic pathway, leading to insulin resistance [39,40]. At the same time, increased inflammation inhibits DIO1 and DIO2 and induces DIO3, which impairs T4-to-T3 transformation and decreases T3 bioavailability, turning the TH-dependent signaling pathway from a classical, nuclear pathway to alternative, nongenomic pathways [43,60]. This may also serve as a compensatory mechanism for developed insulin resistance by bypassing suppression of insulin-mediated activation of the PI3K/Akt pathway. In these terms, the nongenomic TH-dependent signaling pathway was found to activate Akt-kinases, but in parallel, it also activates the stress-related MAPKs, which augments ROS production and upregulation of HIF-α1, all leading to activation of the NLRP3 inflammasome [43,60]. The tissue-related anti-inflammatory and homeostatic mechanisms, notably those associated with IL-37, come into play and intervene in this vicious cycle of increasing levels of inflammation and impairment in glucose-related metabolism [106]. The extraglandular production of TSH in activated immune and residential tissue cells is likely to be a part of these compensatory mechanisms, aiming to reverse the impaired oxidative metabolism [58]. Increased production of insulin in pancreatic β-cells, as a response to increasing insulin resistance, may function in the same way [107].

When tissue-related compensatory mechanisms become insufficient or when inflammation progresses, an increase in insulin resistance at the global (whole-body) level becomes a stimulus for HPT axis activation, as a system-level compensation for the failure of THs to achieve metabolic effects in tissues. In light of that, a slight increase in serum TSH can be considered a marker of system-level insulin resistance. Further progression of inflammation/insulin resistance leads to a decrease in TSH-dependent TH production as well as reducing sensitivity of tissues to metabolically active THs [119]. In the case of overwhelming inflammation, both tissue-related and system-level compensatory mechanisms tend to become exhausted. Through the action of proinflammatory cytokines on the hypothalamus, the HPT axis becomes disrupted, losing the TSH–fT3 feedback loop [64,118]. NTIS has emerged as a sign of whole body-level breakdown in homeostasis. In line with this hypothesis, results from observational studies indicate that long-lived individuals are usually characterized with discrete low thyroid function, whereas in older individuals with chronic conditions and multimorbidity, the appearance of NTIS is a sign of poor prognosis [118,124]. To state it differently, higher serum TSH levels are likely to reflect insulin resistance and MS state associated with overweight/obesity, whereas NTIS reflects insulin resistance associated with frailty and lower body weight, as in NTIS associated with chronic renal failure [125].

## 7. Markers of Target Organ Damage can Contribute to Risk Stratification of Older Individuals with Metabolic and Vascular Comorbidities

Age-related metabolic and vascular conditions, such as obesity, hypertension, T2D, and CVD, usually overlap, both within an individual and between individuals, as an expression of the common pathophysiological background that relies on the dynamic interplay between metabolic, inflammatory, and vascular pathways [17,126]. The efforts to apply personalized prevention and treatment strategies in older individuals expressing these disorders are compromised by a lack of knowledge of how modifying factors, including both genetics and environmental factors, influence the major pathophysiological pathways and how this relates to clinical expression of these disorders [127]. The fact that metabolic and vascular conditions appear at higher rates in individuals of older age further accentuates the heterogeneity of phenotypes by increasing the potential of these patients to develop multiple comorbidities and geriatric syndromes such as malnutrition, sarcopenia, and frailty [128].

Knowing that age-related metabolic and vascular conditions are systemic inflammatory disorders and accompanied by multiple organ damage demonstrates the need to link a patient’s clinical features with pathophysiological pathways and the level of organ damage [6,8,17]. The processes responsible for organ damage and factors influencing dynamics and intensity of this damage are additional challenging issues that need to be clarified before efficient individualized prediction and management strategies can be established for individuals with these disorders [1,2]. The factors often neglected here as being difficult to measure in real-life settings are factors such as tissue receptivity for new inflammatory/immune cells and the speed and intensity of tissue damage. Expectations are high for applying new systems-biology research approaches and new molecular-biology techniques, such as transcriptomic or imaging mass cytometry analyses at the single-cell level, which are thought as to be able to provide insights into the composition and connectivity of cells in tissue and organs [129,130].

Current understanding of the processes that remain in the background of organ damage progression is that it is an unresolved inflammatory reaction, with the bias oscillating between more accentuated inflammatory or reparative processes [1,2]. As recent evidence suggests, in addition to macrophages, the adaptive immune mechanisms, especially the skewed balance between the proinflammatory Th17 and anti-inflammatory Treg subsets, also play a role in creating a proinflammatory environment and in the organ damage under these conditions, similar to that in autoimmune diseases [29,30,31,32,131]. However, due to the unresolved tension between Th17 and Treg cells, this leads to both in situ expansion of Treg cells and a continuous supply of tissue with neutrophils [29]. These two cell lines are the major source of tissue fibrotic and remodeling factors owing to their ability for increased production of ROS (neutrophils) and TGF- β (Treg cells) [132,133]. In addition to the cytokine IL-17 being a measurable marker of tissue inflammation and damage, IL-37 has recently emerged as another, and even more comprehensive, measurable marker of tissue stress in age-related metabolic and vascular conditions [134,135].

## 8. The Model of Target Organ Damage in Older Patients with T2D Based on Patterns of Serum TSH and IL-37

By keeping the aforementioned facts in mind and based on the recent evidence indicating that factors such as an individual’s age and the age of hypertension and T2D duration and the age of onset may influence the rates of CV risk factor accumulation and organ damage progression, we designed a simple cluster analysis for the group of older individuals diagnosed with T2D [136,137]. In creating this study design, we were guided by the recent trend in research on T2D, based on using clustering techniques in determining the phenotypic subgroups of these patients [138,139], as well as by our experience in using these techniques in geriatric studies [140,141]. As markers of target organ damage, we included the cytokines IL-17 and IL-37 [142]. The results indicated that there are strong correlations between these two cytokines and between the cytokine IL-37 and the hormone TSH. In addition, variations among the identified phenotypic subgroups in patterns of IL-17, IL-37, and TSH were shown to reflect differences in the presence of CV complications and the potential of individuals in the clusters for inflammatory/immune cell recruitment and organ damage progression.

We hypothesized that a slight increase in TSH can compensate for, and be a marker of, the whole body‘s insulin resistance, reflecting system-level inflammation, and that variations in IL-37 indicate the total level of tissue stress, which, in addition to inflammation-mediated metabolic changes, also involves inflammation-mediated tissue damage (Table 1). Levels of innate immune inflammation, which is known to underlie insulin resistance, and differences in the degree of tissue/organ damage, may both vary among individuals, but both adaptive immune mechanisms and intrinsic, tissue-dependent controlling mechanisms also play a role here. Variations in the patterns of TSH and IL-37 among individuals in the clusters may reflect the shift in these two levels of stress signaling (Table 1). Even in part, interindividual differences in the progression of organ damage and rates of aging may be due to gene polymorphism for the suppressor cytokine IL-37, or some other regulatory molecules, such as SIRT1, which may account for the variability in longevity in an older population [113,121]. We believe that this simple model, indicating the shift in the tissue/organ level stress response and the whole body’s stress response, if further elaborated, and by involving new, simple, measurable biomarkers, could revolutionize the personalized approach to managing age-related metabolic and vascular conditions.

## 9. Conclusions

In this work, we considered the need to create new types of models for risk stratification of older individuals with T2D (or cardiometabolic comorbidities in general). This viewpoint emerged from several pieces of evidence: (1) recommendations of recent guidelines to include comorbidities and information on target organ damage into risk prediction models for older individuals with cardiometabolic conditions, (2) new findings about mechanisms that underlie chronic inflammation in target organ damage, (3) new findings indicating the role of tissue-related defense mechanisms in determining the dynamics of target organ damage, and (4) a new trend in research based on using new methodologies (concretely, clustering techniques) to identify phenotypes of older patients with complex chronic diseases, such as T2D, which would not be possible without using these methods. We suggest a model of phenotypes of older individuals with T2D that is based on clustering IL-37 and TSH as markers of the tissue level and system level of protective mechanisms that can compensate for inflammation-related organ damage and associated metabolic costs. The main limitations of this model are the small number of participants, the lack of participants in poor health (all patients were independently mobile), and the lack of information about the serum levels of THs (fT4 and fT3) to indicate the presence of NTIS. Nevertheless, we believe that this work will initiate a new wave of research aimed at searching for new biomarkers and new types of models to predict health-related outcomes under aging conditions.

## Figures and Tables

**Figure 1 ijms-23-06456-f001:**
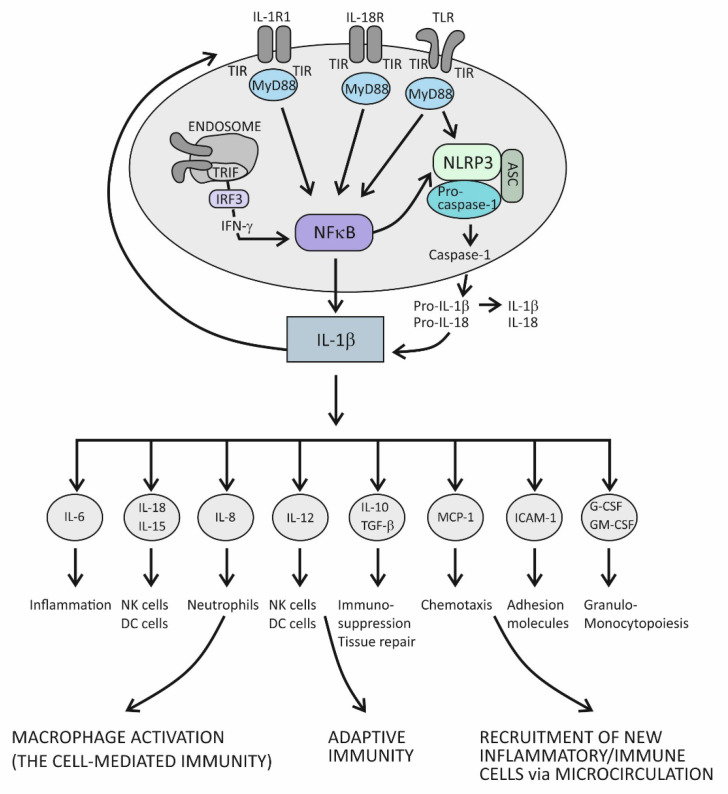
The role of the IL-1 cytokine family in promoting inflammation.

**Figure 2 ijms-23-06456-f002:**
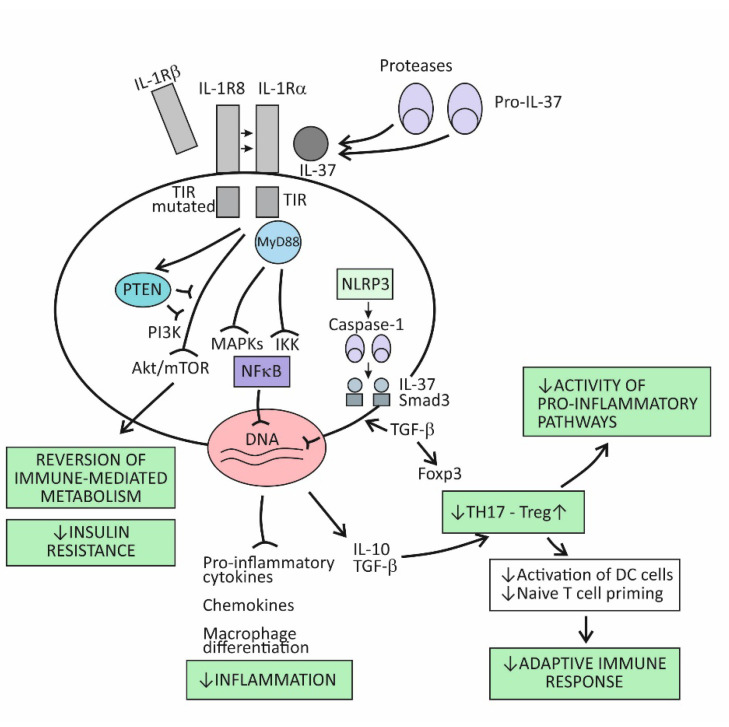
The immunosuppressive and metabolic effects of the cytokine IL-37.

**Figure 3 ijms-23-06456-f003:**
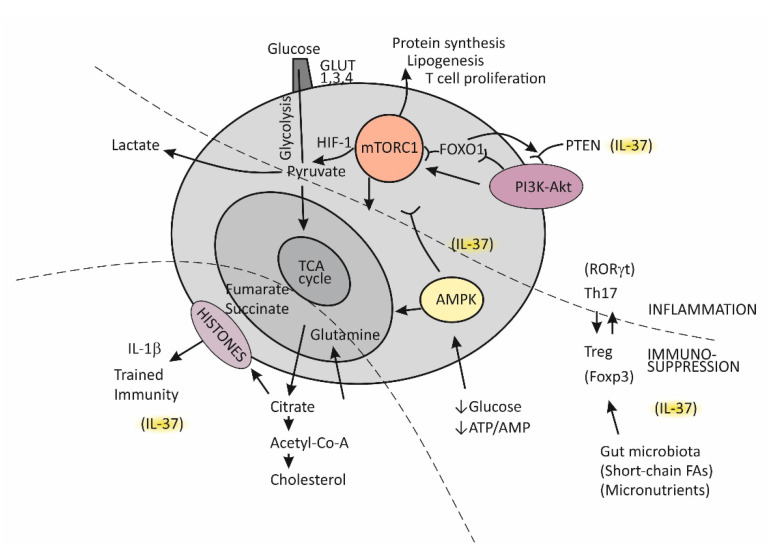
Metabolic reprograming in chronic inflammation and the balance between the tolerogenic and inflammatory immune axis.

**Table 1 ijms-23-06456-t001:** The patterns of serum TSH and IL-37 levels in revealing patterns of target organ-related inflammation and system-level inflammation in age-related metabolic and vascular conditions, according to Ref. [142].

Phenotype	Inflammation- and Metabolic-Related Changes	Patterns of Serum TSH (Min–Max 0.15–9.45) (Mean 2.91) (mU/L) and IL-37 (Min–Max 0.14–258.80) (one extreme value excluded), IQR (21.2–48.0) (Median 13.4)
Slow-rate progression of organ damage Low-level system Inflammation—low-level insulin resistance	Low-rate migration of inflammatory/immune cells into tissues Low stimulus for IL-37 induction Low-level insulin resistance	Low IL-37 (0.24) TSH normal (2.1)
Rapid-rate progression of organ damage but low-level organ damage Low-level system inflammation—low-level insulin resistance	Rapid migration of inflammatory/ immune cells into tissue Moderate tissue inflammation overcomes tissue damage Inflammation/metabolic changes maintained at the tissue level	Higher IL-37 (10.2)—can compensate for metabolic cost of tissue inflammation TSH normal (2.2)
Slow-rate progression of organ damage Medium-level organ damage Higher-level system inflammation—higher-level insulin resistance	Low rate of inflammatory/immune cell migration into tissues Tissue repair processes succeed to reverse tissue homeostasis Low stimulus for IL-37 induction The need for TSH to overcome metabolic costs of increased system inflammation	Low IL-37 (0.22) TSH higher (2.39)
High-level organ damage High-level system inflammation—high-level insulin resistance	Low-rate inflammatory/immune cell migration into tissues—low tissue receptivity for new cells High-level tissue inflammation— high-level organ damage (fibrosis)— repair and remodeling processes Balanced with inflammatory processes	High IL-37 (16.4) (the need to ameliorate tissue fibrotic processes and to compensate for metabolic costs of inflammation) (1) TSH high (3.3) (the need to compensate for high-level insulin resistance) (2) TSH normal (hypothetically) (tissue resistance to thyroid hormones—the HPT axis is broken down)
Moderate-level organ damage Low-level system inflammation—low-level insulin resistance	Low-rate inflammatory/immune cell migration into tissues Moderate-level organ damage— slow-rate repair processes (tissue homeostasis reversed)	Slightly higher IL-37 (0.8) TSH normal (1.9)
Low-level organ damage Low-level system inflammation—low-level insulin resistance	Moderate-to-rapid rate of inflammatory/immune cell migration into tissues Low-level organ damage—the need for repair processes to take place Increasing stimulus for IL-37 induction	Higher IL-37 (3.4) TSH normal (2.2)

## Data Availability

Not applicable.

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
