# Peer review of "Cross-Talk between the Cytokine IL-37 and Thyroid Hormones in Modulating Chronic Inflammation Associated with Target Organ Damage in Age-Related Metabolic and Vascular Conditions"

_ijms, 2022, doi:10.3390/ijms23126456_

Round 1

Reviewer 1 Report

In my opinion the relation of chronic inflamation with metabolomic and vascular changes is very well presented by Trtica Majnaric et al. Therefore, the reviewed text required only minor recommendations.

180 references is an impressive number, the ones from 2004 and 2006 seem to be quite old. So if there are no significant due to the rest of the references I advise removing them. 

p.2 "large range" may be better will be "wide range".

There is no need to present an abbreviation if it is not used further in the text e.g. Advanced glycation end products- AGEs, DAG etc.

p.3 there is also diabetic dermopathy, also an interesting and important symptom of diabetic complications.

Once the Authors have written T2D and D2T, in my opinion, T2D is the correct option.

"via" could be written in italics "via", as well as "in vitro" (e.g. page 17). In several places in the text there is a double space between words. 

Chapter 6 has too long title, the Authors should shorten it.

Table 1. should contain any data in number for example the content of IL-37 and TSH from cited literature. 

7. the begging could be "In this eleborated manuscript..."

Author Response

Dear Editor and the Reviewer No. 1,

Thank you for giving us a good guide for making a revision of the manuscript.

Overall, we have prepared the paper revision according to your comments. This included the following changes:

  • we have reduced the number of references
  • we have changed the term „large range" in "wide range"
  • we have added a reference associated with diabetic dermopathy (chapter 2)
  • we have used the term T2D as the correct option in further text.
  • we have reduced and changed the title of chapter 6 in „ The cross-talk between the cytokine IL-37 and thyroid hormones – a relationship between tissue-related and the whole body stress responses „ and according to comments of reviews 2- revised and comprehend chapter 6 and rewrote two new chapter 7 and 8,  in order to better explain the hypothesis of cross-talk between the cytokine IL-37 and thyroid hormones- with more examples and more support data
  • we made minor corrections e.g double space between words.
  • we made the mentioned corrections in Table 1.

We hope that we have now met all of your expectations regarding the correctness of our manuscript and that there are no further barriers for its acceptance for publication.

Zvonimir Bosnic

Reviewer 2 Report

The manuscript entitled: “The cross-talk between the cytokine IL-37 and thyroid hor-mones in modulating chronic inflammation associated with tar-get organ damage in age-related metabolic and vascular condi-tions”is a review manuscript wich gives a current snapshot on chronic inflammation in the framework of age-related metabolic and vascular conditions. The manuscript fits scope and aims of the Journal. There are some drawbacks.

The hypothesis of cross-talk between the cytokine IL-37 and thyroid hormones should be better assessed in the manuscript as well as the limits and possible other effects. The end points of the manuscript, e.g. the association and role between cytokine IL-37 and thyroid hormones in the mechanism of chronic inflammation associated with organ damage and age should be better detailed and cleared in the text substantating also with examples and more support data.

Nonetheless, one drawback is that the main end point (“the cross-talk….”) as per the title manuscript is detailed only at page 18 in section 6 of the manuscript after a long even if interesting report of data.

Some minor comments are detailed in the following.

Please be consistent wen using shortening in the context, e.g. “CD” for cardiovascular: please define in full at their first use any abbreviation.

In the Table please add also the appropriate relevant substantiating References.

The Conclusion section should be more clear and address better the end points and limits of the proposed “elaborative paper”. This term also should be replaced with “perspective” or review”, considering the perspective point of view given by the Authors. This section should be rephrased.

The English writing should be checked and assessed better for easier readability.

Author Response

Dear Editor and the Reviewer No. 2,

Thank you for giving us a good guide for making a revision of the manuscript. We have prepared the paper revision according to your comments. This included the following changes:

  • we have reorganised prior chapter 6, rewrote, revised and comprehend chapter and rewrote two new chapter 7 and 8, in order to better explain the hypothesis of cross-talk between the cytokine IL-37 and thyroid hormones- with more examples and more supported data
  • the association and role between cytokine IL-37 and thyroid hormones in the mechanism of chronic inflammation associated with organ damage and age has been explained in chapter 8
  • we have reduced and revised chapter 3
  • we have defined in full any abbreviation at their first use in manuscript
  • we made the mentioned corrections in Table 1.
  • we have added and better explained limits and possible other effects in conclusion section
  • the end points of the manuscript and conclusion section has been rephrased
  • we also made minor corrections in manuscript
  • manuscript was proofread by English expert

In addition to these changes , we hope that we have now met all of your expectations regarding the correctness of our manuscript and that there are no further barriers for its acceptance for publication.

Round 2

Reviewer 2 Report

The manuscript has been modified, nonetheless there are still points to clear. The manuscript is still not easy to follow in the overall reasoning and end points and scope. There is still a very long part divided in sections before getting to the main end point of the manuscript discussed in section 6 after 20 pages.This should be the main topic of the manuscript and commented before in previous sections which could be reduced.

 In the Table 1 please add the appropriate substantiating relevant References. In section 8 of the manuscript the Authors mention a “cluster analysis” developed by them: please add more information in this regard.

In the Conclusion section, the sentence: “The main limitations of this model are the small number of participants…..” should be substantiated by appropriate References.

Author Response

Dear Editor,

Regarding the last objections of one of the reviewers on the revised version of our manuscript entitled: “The cross-talk between the cytokine IL-37 and thyroid hormones in modulating chronic inflammation associated with target organ damage in age-related metabolic and vascular conditions”, we would like to provide the following explanations:

In this manuscript, we defend the hypothesis that has arisen from our preliminary work, cited in the ref. 144. We marked this ref. besides the title of the Table 1. There is lot of recent evidence to make a background for creating this hypothesis, but scattered between different areas of knowledge. We aimed to clearly represent these different pieces of evidence and integrate them into the common picture.

The added chapter 6, as a cornerstone of the article, integrates these already described particular concepts. It logically follows after the previous parts of the article. Also, the suggestion of the second reviewer to reduce many sections is not acceptable for us. We have already invested much time in preparing this article, and made a major reconstruction in the first round of the revision process. We have proofread the manuscript, and did the best we knew. The article has been edited by the native English language speaker. The decision is up to you.

We only made some corrections in the literature citations (marked in yellow color), as the last time, there was the need to cut out some parts of the text and insert them into other places, and we had to changes the places for almost all references.

Ljiljana Trtica Majnarić, assoc. prof.
